# In-Situ Metatranscriptomic Analyses Reveal the Metabolic Flexibility of the Thermophilic Anoxygenic Photosynthetic Bacterium *Chloroflexus aggregans* in a Hot Spring Cyanobacteria-Dominated Microbial Mat

**DOI:** 10.3390/microorganisms9030652

**Published:** 2021-03-21

**Authors:** Shigeru Kawai, Joval N. Martinez, Mads Lichtenberg, Erik Trampe, Michael Kühl, Marcus Tank, Shin Haruta, Arisa Nishihara, Satoshi Hanada, Vera Thiel

**Affiliations:** 1Department of Biological Sciences, Tokyo Metropolitan University, Hachioji, Tokyo 192-0397, Japan; j.martinez@usls.edu.ph (J.N.M.); mat19@dsmz.de (M.T.); sharuta@tmu.ac.jp (S.H.); arisa.nishihara@aist.go.jp (A.N.); satohana@tmu.ac.jp (S.H.); 2Institute for Extra-cutting-edge Science and Technology Avant-garde Research (X-star), Japan Agency for Marine-Earth Science and Technology (JAMSTEC), Yokosuka, Kanagawa 237-0061, Japan; 3Department of Natural Sciences, College of Arts and Sciences, University of St. La Salle, Bacolod City, Negros Occidental 6100, Philippines; 4Department of Biology, Marine Biological Section, University of Copenhagen, Strandpromenaden 5, 3000 Helsingør, Denmark; mlichtenberg@sund.ku.dk (M.L.); etrampe@bio.ku.dk (E.T.); mkuhl@bio.ku.dk (M.K.); 5DSMZ—German Culture Collection of Microorganisms and Cell Culture, GmbH Inhoffenstraße 7B, 38124 Braunschweig, Germany; 6Bioproduction Research Institute, National Institute of Advanced Industrial Science and Technology (AIST), Ibaraki 305-8566, Japan

**Keywords:** filamentous anoxygenic phototroph, microbial mats, hot springs, metatranscriptomics, energy metabolism, carbon fixation

## Abstract

*Chloroflexus aggregans* is a metabolically versatile, thermophilic, anoxygenic phototrophic member of the phylum *Chloroflexota* (formerly *Chloroflexi*), which can grow photoheterotrophically, photoautotrophically, chemoheterotrophically, and chemoautotrophically. In hot spring-associated microbial mats, *C. aggregans* co-exists with oxygenic cyanobacteria under dynamic micro-environmental conditions. To elucidate the predominant growth modes of *C. aggregans*, relative transcription levels of energy metabolism- and CO_2_ fixation-related genes were studied in Nakabusa Hot Springs microbial mats over a diel cycle and correlated with microscale in situ measurements of O_2_ and light. Metatranscriptomic analyses indicated two periods with different modes of energy metabolism of *C. aggregans*: (1) phototrophy around midday and (2) chemotrophy in the early morning hours. During midday, *C. aggregans* mainly employed photoheterotrophy when the microbial mats were hyperoxic (400–800 µmol L^−1^ O_2_). In the early morning hours, relative transcription peaks of genes encoding uptake hydrogenase, key enzymes for carbon fixation, respiratory complexes as well as enzymes for TCA cycle and acetate uptake suggest an aerobic chemomixotrophic lifestyle. This is the first in situ study of the versatile energy metabolism of *C. aggregans* based on gene transcription patterns. The results provide novel insights into the metabolic flexibility of these filamentous anoxygenic phototrophs that thrive under dynamic environmental conditions.

## 1. Introduction

Members of the genus *Chloroflexus* are thermophilic, filamentous anoxygenic phototrophs (FAPs) in the phylum *Chloroflexota* (formerly *Chloroflexi*). They are well known to have the ability to grow photoheterotrophically under anaerobic conditions and chemoheterotrophically under aerobic conditions in the laboratory [1,2,3]. While photoautotrophic growth in the laboratory has been observed only in a small number of isolated strains (e.g., *Chloroflexus aurantiacus* strain OK-70-fl [4,5,6], *Chloroflexus* sp. strain MS-G [7], and *Chloroflexus aggregans* strains NA9-6 [8,9] and ACA-12 [10]), the genes necessary for the 3-hydroxypropionate (3-OHP) bi-cycle, which is a carbon fixation pathway found only in members of the order *Chloroflexales* among bacteria, are present in all of the available *Chloroflexus* spp. genomes [11]. *C. aggregans* strains NA9-6 and ACA-12, which were isolated from Nakabusa Hot Springs (Nagano Prefecture, Japan), can grow photoautotrophically with hydrogen gas (H_2_) [8] and sulfide [10] as the electron donors in pure culture, respectively. In addition to the long known phototrophic and chemoheterotrophic metabolism in *Chloroflexus* spp., chemoautotrophic growth has recently been shown in lab studies of *C. aggregans* strain NA9-6 [8]. In addition, fermentative growth has been shown in two isolates of *C. aurantiacus*, strains B3 and UZ [12].

Microbial mats in the slightly alkaline, sulfidic Nakabusa Hot Springs have been intensively studied with regard to their microbial diversity and functions [13,14,15,16,17,18,19,20,21]. At water temperatures of 63–70 °C, olive-green microbial mats (“*Chloroflexus* mats”) are dominated by *C. aggregans* [14,15], and oxygenic cyanobacteria are not found. At lower temperatures of 45–62°C, *Chloroflexus* spp. co-exist with cyanobacteria in dark blue-green microbial mats (“cyanobacterial mats”). These blue-green mats are stratified with a green upper layer dominated by the thermophilic cyanobacteria on top of an orange-colored layer that is frequently inhabited by *C. aggregans* [13].

In the anoxygenic, cyanobacteria-free phototrophic mats, *C. aggregans* is considered to be the main primary producer, using sulfide as the major electron source [9,10,14,15,16]. The metabolic repertoire of *C. aggregans* in the blue-green cyanobacterial mats has remained unstudied. In situ isotopic studies of similar cyanobacterial mats colonizing the effluent channels of Mushroom Spring and Octopus Spring in Yellowstone National Park (YNP; WY, U.S.) suggested that filamentous phototrophic *Chloroflexota* vary their carbon metabolisms over a diel cycle [22]. Based on transcriptomic data, Klatt et al. (2013) suggested photomixotrophic growth of a member of FAPs—i.e., *Roseiflexus* spp.—in Mushroom Spring during daytime and fermentative growth during the night [23]. Compared to the microbial mats in YNP, Nakabusa Hot Spring cyanobacterial mats are rich in *C. aggregans*, at a relative abundance of approximately 21–22% compared to only 1% in Mushroom Spring cyanobacterial mats [13,24]. This suggests an important ecological role and potential function of *C. aggregans* as a primary producer in the Nakabusa mats.

In this study, the in situ metabolic lifestyle of *C. aggregans* in the blue-green microbial mats of Nakabusa Hot Springs was analyzed by using a metatranscriptomic approach. Light is the main energy source during daytime, supporting photoautotrophic, photomixotrophic and photoheterotrophic growth of *C. aggregans*, while chemotrophic growth is prevalent during the afternoon and night. During the afternoon, under microaerobic low-light conditions chemoheterotrophic growth is based on O_2_ respiration, while at night fermentation is conducted under anaerobic conditions. Unexpectedly, chemoautotrophic growth using O_2_ as the terminal electron acceptor appeared to take place during early morning hours before sunrise, which suggests a vertical migration of *C. aggregans* cells to the microaerobic surface layers of the mats.

## 2. Materials and Methods

### 2.1. Field Site and Sample Collection

Blue-green cyanobacterial mat samples were collected from a small pool at 56 °C with slightly alkaline (pH 8.5–8.9) and sulfidic (46–138 µM) hot spring water [18,25,26,27] at Nakabusa Hot Springs, Nagano Prefecture, Japan (36°23′33″ N, 137°44′52″ E) [20]. Microbial mat samples of approximately 3 mm thickness with two distinct vertical layers, a green top layer and an orange-colored bottom layer (Figure 1), were randomly collected in triplicate using a size 4 cork borer (8 mm diameter) as previously described [13,14]. Samples were placed in 2 mL screw-cap tubes and snap-frozen in a dry-ice cooled, 70% (v/v) ethanol bath on site. Samples were taken at 12 different time points over a diel cycle on 3 to 4 November, 2016 (19:00 and 23:00 on 3 November; 02:10, 05:00, 06:00, 07:00, 11:00, 15:00, 16:00, 17:00, 18:00, and 19:00 on 4 November) and were brought back to the laboratory on dry ice and stored in a −80 °C freezer until further processing for metatranscriptomic analyses.

### 2.2. RNA Extraction

RNA extraction from microbial mat samples was performed as previously described [18]. Briefly, 0.10–0.21 g wet weight samples were used for RNA extraction with an RNeasy PowerBiofilm Kit (Qiagen, Valencia, CA, USA) following the manufacturer’s protocol. The RNA was treated with DNase I and eluted with RNase-free water. Purity and concentration of the RNA were determined using an RNA High Sensitivity (HS) assay with a Qubit 3.0 fluorometer (Life Technologies, Grand Island, NY, USA).

### 2.3. RNA Sequencing

Library preparation and sequencing of the RNA samples were conducted at DNALink Inc. (Seoul, Korea) as described previously [18]. RNA purity was determined by assaying 1 µL of total RNA extract on a NanoDrop8000 spectrophotometer (Thermo Fisher Scientific, Waltham, MA, USA). Total RNA integrity was assessed by the RNA integrity number (RIN) using a 2100 Bioanalyzer (Agilent Technologies, Palo Alto, CA, USA). Total RNA sequencing libraries were prepared using a Truseq Stranded Total RNA Library prep kit and Ribo-Zero bacteria kit (both from Illumina, San Diego, CA, USA) according to the manufacturer’s instructions.

First, 0.5 µg of total RNA was subjected to ribosomal RNA depletion with Ribo-Zero bacteria reagent using biotinylated probes that selectively bind rRNA species. Following purification, the rRNA-depleted total RNA was fragmented into small pieces using divalent cations under elevated temperature. The cleaved RNA fragments were copied into first-strand cDNA using random primers and reverse transcriptase, followed by second-strand cDNA synthesis using DNA polymerase I and RNase H. A single ‘A’ base was then added to these cDNA fragments, and the adapter was ligated. The products were purified and enriched by polymerase chain reaction (PCR) to create the final cDNA library.

The quality of the amplified libraries was verified by capillary electrophoresis using the 2100 Bioanalyzer (Agilent Technologies, Palo Alto, CA, USA). After a quantitative (q)PCR using SYBR Green PCR Master Mix (Applied Biosystems, Carlsbad, CA, USA), index-tagged libraries were combined in equimolar amounts. RNA sequencing was performed using an Illumina NextSeq 500 system following the provided protocols for 2 × 150 sequencing.

### 2.4. Sequence Data Analyses

Raw RNA reads were pre-processed using FastQC [28]. Adapter sequence and low-quality reads were trimmed by Cutadapt ver. 1.12 [29]. Quality-checked reads were mapped against the complete genome of *C. aggregans* DSM 9485^T^ (RefSeq acc. No. NC_011831.1) [2] with bowtie2 ver. 2.3.0 [30] with default settings allowing no mismatches. The reads were then aligned using the EDGE-pro algorithm [31] with the rRNA depletion option.

Transcriptomic analyses were conducted as described previously [18]. In short, read counts were normalized for each time point by the total number of reads retrieved for the target organism. The relative transcription of each gene during the cycle was then calculated and normalized against the mean of all of the reads at each time point for that particular gene over the diel cycle. This method allows comparison of the relative transcription abundance levels (rather than the absolute values) for each gene across the diel cycle.

### 2.5. Statistical Analyses

As described above, diel transcriptomic data in this study lacked replication. In the following Results and Discussion sections, the authors carefully interpreted and described and intentionally averaged the normalized transcriptional patterns of several genes related in a single pathway to recognize those gene transcription patterns as the pathway-level metabolic dynamics. However, some important genes function in an important enzyme reaction solely, the statistical analyses of each gene were performed to discuss the transcriptional changes over a diel cycle. For each gene in dual datasets, and for every possible pair-wise comparison of the 11 sets of adjacent samples (November 3 19:00–23:00; 3 November 23:00–4 November 2:10; 4 November 02:10–05:00, 05:00–06:00, 06:00–07:00, 07:00–11:00, 11:00–15:00, 15:00–16:00, 16:00–17:00, 17:00–18:00, 18:00–19:00), “exactTest” program in edgeR with dispersion set at 0.1 was used to determine the probability that the gene was differentially transcribed in a statistically significant manner [32,33]. 

### 2.6. Microsensor Analyses

The profiles of the O_2_ concentration as a function of depth in the microbial mat were measured in situ by using a Clark-type O_2_ microsensor (OX25; Unisense, Aarhus, Denmark) with a tip diameter of <25 µm, low stirring sensitivity (<1–2%) and fast response time (t_90_ < 0.5 s). The O_2_ microsensor was mounted on a motorized micromanipulator (Unisense, Aarhus, Denmark) and connected to a PC-interfaced pA-meter (Unisense, Aarhus, Denmark), both of which were controlled by dedicated data acquisition, profiling, and positioning software (SensorTrace Pro, Unisense, Aarhus, Denmark). The micromanipulator was mounted on a metal stand placed next to the hot spring, allowing for vertical insertion of the microsensor tip into the microbial mat under natural flow, temperature and light conditions. The microsensor tip was carefully positioned at the mat surface (defined as 0 µm) by manual operation of the micromanipulator. Subsequently, O_2_ microprofiles were recorded automatically every 15 min for 24 h starting at 18:00 on 3 November 2016. In each profile, O_2_ measurements were made in 100 µm increments from the water-phase and into the mat. One measurement was taken per depth and, for each measurement a 10 s wait period was applied, to ensure steady O_2_ signal, and the O_2_ signal was then recorded averaged over a 1 s period.

### 2.7. Irradiance Measurements

Downwelling solar photon irradiance (400–700 nm) at the water surface next to the mat was logged every 5 min throughout the 24 h diel sampling cycle with a calibrated light meter connected to a cosine-corrected photon irradiance sensor (ULM-500, MQS-B; Walz, Effeltrich, Germany).

## 3. Results

### 3.1. Irradiance and In Situ Oxygen Dynamics in the Microbial Mat

The O_2_ concentration and penetration from the surface green layer to the deeper orange layer in the microbial mat varied dramatically with irradiance. The whole mat was anoxic during the night, and the O_2_ concentration started to increase at the mat surface at around 06:00, correlating with the time of sunrise and thus the onset of cyanobacterial oxygenic photosynthesis under diffuse light (Figure 1 and Figure 2). However, O_2_ did not accumulate in deeper mat layers until later in the morning at around 09:00, when the microbial mats were exposed to direct sunlight as the sun rose over the surrounding mountains. Supersaturating O_2_ levels were observed in the uppermost mat layers at ~12:00 (noon) during the highest solar irradiance (1531 µmol photons m^−2^ s^−1^; 400–700 nm). The maximum O_2_ concentration at the microbial mat surface reached >900 µmol O_2_ L^−1^ and with a maximal O_2_ penetration of >2 mm depth under the highest irradiance between 10:00 and 14:00. Thus, there was sufficient O_2_ available for aerobic microbes in the upper 2 mm of the mat during this period of the day. In the afternoon, after 14:00, O_2_ concentration started to decrease gradually (from approx. 500 µmol O_2_ L^−1^) and no O_2_ was detected at a depth of 1–2 mm shortly after 15:00. The upper layer remained oxic (100–200 µmol O_2_ L^−1^) until 16:00. At this time, the microbial mats experienced a substantial decrease in solar irradiance (see Figure 3, Figure 4, Figure 5, Figure 6, Figure 7 and Figure 8 and S1–S5) as the sun set behind the mountains. However, low levels of diffuse sunlight (<100 µmol photons m^−2^ s^−1^; 400–700 nm) hit the microbial mats until complete darkness was observed at 17:00. Anoxic conditions started to become established in the lower parts of the microbial mats ~2 h before sunset, potentially enabling anaerobic and microaerophilic metabolism under low-light conditions during this time interval.

### 3.2. Transcriptome Profiles and Differentially Transcribed Genes

Approx. 0.26–5.23 million reads of transcripts were assigned to the *C. aggregans* genome throughout the day (Appendix A). Among 3848 CDSs contained in the genome of *C. aggregans* DSM9485^T^, the number of genes in which more than 10 transcripts were detected in all timepoints was 2542–3506 genes. Statistical significance of transcription level changes of each gene was determined using the *p*-value (*p* < 0.05) based on the “exactTest” function in edgeR [32]. Thousands of genes were differentially transcribed during the period from 19:00 on 3 November to 15:00 on 4 November indicating a versatile and changing transcriptional activity between the different time points. During 15:00–16:00 as well as 17:00–19:00, the numbers of significantly differentially transcribed genes were considerably lower (less than 10% of all CDSs), indicating transcriptional activity of *C. aggregans* during this period was relatively stable compared with former 20 h of the day. Those times were taken together as ‘afternoon’ and ‘evening’, respectively, in which both transcriptional activity as well as environmental conditions are expected to be rather similar. Overall the statistics support the changes in relative transcriptional levels and activity of *C. aggregans* to be significant. Detailed information on the differential transcription of the genes discussed is given in Appendix A.

### 3.3. Transcription of Photosynthesis-Related Genes

*Chloroflexus aggregans* contains a type 2 photosynthetic reaction center complex (RC) and light-harvesting chlorosomes; the main photosynthetic pigments are bacteriochlorophylls (BChls) *c* and *a* [2]. Transcripts of *pufLMC* genes (Cagg_1639–1640 and Cagg_2631) encoding RC proteins in *C. aggregans* showed significant nocturnal patterns and were most abundant in the evening and at night (Figure 3). Similarly, nocturnal transcriptional patterns of chlorosome proteins encoded by *csmAMNOPY* genes (Cagg_1222, Cagg_1209, Cagg_1208, Cagg_2486, Cagg_1206, and Cagg_1296) were detected (Figure 3).

BChl synthesis-related *bch* genes in *C. aggregans* in the mats showed inconsistent transcription patterns, with the highest relative transcription levels either under oxic conditions during the daytime (11:00) and/or under microoxic conditions in the afternoon (15:00; Appendix A). In total, four different groups of transcription patterns could be distinguished for *bch* genes. Group I (*bchH*-III, *bchI*-I and II, *bchY*, *bchZ*, *acsF*, *bchL*, *bchB*, *bchN*, *bchM*) showed highest relative transcription during the daytime (midday, 11:00). Group II (*bchH*-I and II, *bchJ*, *bchF*, *bchX*, *bchC*) showed maximal values in the afternoon around 15:00, with more or less pronounced lows at 11:00. The two other groups showed highest relative transcription levels under anaerobic conditions. Four genes (*bchG*, *bchK*, *bchU*, and *bchP*) showed peaks at 18:00. *bchD* and *bchI*-III had maximal relative transcription in the early morning at 05:00.

Paralogs with different patterns were present for the genes encoding Mg-chelatase, i.e., *bchH* (three paralogs; Cagg_0239, 0575, 1286) and *bchI* (three paralogs; Cagg_1192, 2319, 3123). In all three cases, the paralogs showed differences in absolute read abundance as well as different temporal peaks in relative transcription over the day (Appendix A).

Two different genes encoding for Mg-protoporphyrin monomethylester cyclase are present in *C. aggregans*. AcsF and BchE both catalyze the synthesis of divinylprotochlorophyllide from Mg-protoporphyrin IX 13-monomethyl ester (one of the intermediates in the BChl synthesis pathway) under aerobic and anaerobic conditions, respectively [34,35,36]. *acsF* (Cagg_1285) was transcribed throughout the diel cycle with relative transcription levels about four times higher during day, when the mat was (super)oxic than under anoxic conditions at night (Figure 3). In contrast to *acsF*, the *bchE* (Cagg_0316) showed significantly lower relative transcription levels (only 1/32 of the diel average) under high-light/high-O_2_ conditions in the mat and higher transcription levels during the night as well as during low-light transition times in the morning and afternoon (Figure 3).

### 3.4. Phototrophic and Respiratory Electron Transport

Electron transport chains are involved in both phototrophic and chemotrophic (respiratory) metabolism. *Chloroflexus* spp. contain paralogs of some of the major enzyme complexes involved in the electron transport chain, namely, NADH:menaquinone oxidoreductase (Complex I) [11,37,38] alternative complex III (ACIII) [39,40,41] and the soluble electron carrier auracyanin [42,43,44,45] in their genomes. Single-copy genes are present encoding succinate dehydrogenase (Complex II) [46,47] and F-type ATP synthase (Complex V) [48].

The respiratory complex I carries out the transfer of electrons between soluble cytoplasmic electron carriers and membrane-bound electron carriers coupled to proton translocation generating a transmembrane proton motive force. *C*. *aggregans* DSM9485^T^ contains two sets of genes encoding Complex I (NADH:menaquinone oxidoreductase, *nuo*): one (Cagg_1620–1631) represents a cluster comprising 12 genes with an additional *nuoM* (2-M complex) inserted between the original *nuoM1* and *nuoN* genes (*nuoABCDHJKLMMN*, 3); the second is a complete cluster comprising 14 genes (Cagg_1036–1049). The additional proton-pumping subunit NuoM has been speculated to lead to a higher stoichiometry of protons translocated per 2e^−^ reaction cycle [49]. Both sets of *nuo* genes were transcribed and showed the highest relative transcription during the daytime (Appendix A). The 14-gene set showed a transcriptional peak at 11:00 and the 12-gene peak came slightly later (in the afternoon at 15:00–16:00). Both *nuo* gene sets showed a small transcription peak in the early morning at 05:00.

Alternative complex III (ACIII) transfers electrons from menaquinol to water-soluble proteins such as auracyanin, the blue copper electron carrier protein found in *Chloroflexus* spp. [44]. Two types of ACIII have been reported in *Chloroflexus* spp.: Cp, which is thought to be involved in cyclic phototrophic electron transport, and Cr, which is predicted to be related to the reduction of oxygen in respiratory electron transport [11,23,39]. In the present study, clear diel patterns as well as differences between the two sets of genes were observed. The genes encoding Cp showed high relative transcription levels all day with the highest level at 11:00 and a significant decrease in the afternoon as the sunlight vanished. The Cr genes were not highly transcribed under the daytime high O_2_ conditions in the mat, but they significantly increased and exhibited maximal relative transcription levels in the microoxic low-light afternoon hours at 15:00 and 16:00 (Figure 4). Genes for both Cp and Cr were highly transcribed in the early morning at 5:00.

Two homologs encoding the soluble electron carrier protein auracyanin, *aurA* and *aurB*, are present in the genome of *C. aggregans*. The transcriptional profile of *aurA* in *C. aggregans* in the cyanobacterial mats in the present investigation showed patterns similar to those of Cr and other genes involved in the respiratory electron transport chain, with one significant peak in the early morning at 05:00 and another under microoxic and anaerobic conditions in the afternoon and evening (15:00 until 18:00; Figure 4). In contrast, *aurB* was significantly higher transcribed during a high-light period (11:00) as well as at 05:00.

Respiratory Complex IV—i.e., the cytochrome *c* oxidase complex—plays a key role in the reduction of O_2_ to H_2_O in the respiratory electron transport chain. The oxygen profiles over the diel cycle indicated high O_2_ concentrations in the mats during the daytime (Figure 2). In the laboratory, *Chloroflexus* spp. can grow chemoheterotrophically using respiration under aerobic dark conditions. In *Chloroflexus* spp., the cytochrome c oxidase (COX, or Complex IV; EC 1.9.3.1) genes are clustered with the Cr operon (Cagg_1519–1522). Similar to Cr, the average relative transcription levels of these COX genes reached their highest values in the early morning at 05:00 and in the afternoon at 15:00 and 16:00 (Figure 5).

*Chloroflexus aggregans* possesses a type-B succinate dehydrogenase (Complex II) which comprises one polypeptide and two hemes for a transmembrane cytochrome *b* (*sdhC*) in addition to a flavoprotein subunit (*sdhA*) and iron-sulfur subunit (*sdhB*) [50,51]. Complex II encoded by *sdhCAB* (Cagg_1576–1578) is involved in electron transport as well as in the TCA cycle and the 3-OHP bi-cycle. They showed significant high relative transcription levels in the morning at 05:00 and during the daytime, and were significantly low throughout the night (Figure 5).

F-type ATP synthase is involved in the production of ATP based on the proton motive force obtained by phototrophic as well as respiratory electron chain activity. The ATP-synthase consists of two parts: F^1^, which is a catalytic part, and F^0^, which is a transmembrane proton channel part [48]; *C*. *aggregans* DSM9485^T^ contains a complete gene set for both parts in the genome [11]. Similar to Complex I-1, II, and ACIII Cp, the relative transcription of ATP-synthase showed a diurnal pattern with two peaks, one at 05:00 and the other at 11:00 (Figure 5).

### 3.5. 3-Hydroxypropionate Bi-Cycle and Anaplerotic Carbon Fixation

*Chloroflexus* spp. contain all genes for the 3-hydroxypropionate (3-OHP) bi-cycle, a carbon fixation pathway found only in members of filamentous phototrophic *Chloroflexota* [11,52,53,54,55,56,57]. The number of transcripts per million (TPM) for genes encoding key enzymes of the 3-OHP bi-cycle, i.e., malonyl-CoA reductase (Cagg_1256) and 3-hydroxypropionyl-CoA synthase (Cagg_3394) [23], were considerably higher than the average of all genes and appeared relatively stable over the diel cycle (Appendix A). Although transcription was detected at all times, the relative transcripts of the two key enzyme genes on average peaked at 15:00. The second highest peak of 3-OHP bi-cycle key enzyme genes was detected at 05:00 before sunrise, and again at 07:00. Under the oxic, high-light conditions, at 11:00, the relative transcriptional levels of the genes encoding key enzymes of 3-OHP bi-cycle were the lowest (Figure 6).

Filamentous anoxygenic phototrophs also contain anaplerotic pathways for incorporating inorganic carbon, such as phosphoenolpyruvate carboxylase (*ppc*, Cagg_0399), which catalyzes the unidirectional production of oxaloacetate from phosphoenolpyruvate [11,58,59]. Transcripts of *ppc* were abundant during the daytime under aerobic, high-light conditions (Figure 6).

### 3.6. Electron Donors: Hydrogenase, Sulfide: Quinone Reductase and CO-Dehydrogenase

Recent studies demonstrated that *C. aggregans* has the capability to use sulfide as well as H_2_ as an electron donor for photoautotrophic growth [9,10]. Genome analyses have suggested that carbon monoxide can also serve as a potential electron donor [11]. The correlation between the relative transcriptions of genes encoding hydrogenases, sulfide:quinone reductase and carbon monoxide dehydrogenase and the genes encoding key enzymes in the 3-OHP bi-cycle was analyzed in order to predict autotrophic metabolism.

*C. aggregans* contains two Ni-Fe hydrogenases: a bidirectional hydrogenase (Cagg_2476–2480) and an uptake hydrogenase (*hyd*, Cagg_0470–0471)–that can provide electrons for autotrophic growth [60,61]. Relative transcription levels of *hyd* genes encoding the uptake hydrogenase significantly increased in the afternoon (at 15:00) shortly after the direct solar illumination of the mats ended around 14:00 and anaerobic conditions were established in the deeper layers of the mat, as well as in the early morning at 05:00 (Figure 2 and Figure 7). The relative transcription levels for genes encoding the bidirectional hydrogenase (Cagg_2476–2480) peaked a little later in the evening after sunset (17:00, PAR=0), stayed high throughout the night, and then decreased during the day. Nickel transporter genes showed the same high relative transcription pattern as the *hyd* genes, with peaks in the early morning and the afternoon (Figure 7).

*Chloroflexus* spp. contain type-II sulfide:quinone oxidoreductase (SQR), which oxidizes sulfide to elemental sulfur, but lack the *dsr* genes, which encode genes involved in the oxidation of elemental sulfur to sulfate as observed in green and purple sulfur bacteria [11,62], and also lack the *sox* system, which oxidizes elemental sulfur and thiosulfate to sulfate and is widespread in chemoautotrophic sulfur oxidizers [63]. In the present study, a significant increase of relative transcription levels with the highest peak of *sqr* was detected in the afternoon, at 15:00 (Figure 7) under microaerobic to anaerobic low-light conditions in the mat.

As a third possibility, based on the presence of genes encoding carbon monoxide dehydrogenase (*coxGSML*, Cagg_0971–0974) in the genome, the capability of *Chloroflexus* spp. to utilize CO as an electron donor and/or carbon source during aerobic or microaerobic growth has been discussed [11]. In the present investigation, the *coxGSML* genes were significantly higher transcribed during high-light conditions around noon (Figure 7) as well as during the afternoon as the mats turned anoxic. A significant increase in the relative transcription of *cox* genes was seen under anaerobic, low-light conditions in the morning at 07:00 together with a spike in the relative transcription for genes encoding key enzymes of the 3-OHP bi-cycle. The relative transcription of hydrogenase and *sqr* genes were clearly and significantly decreased at that time.

### 3.7. Carbohydrate Metabolism and the TCA Cycle

*Chloroflexus aggregans* contains the gene set for the pentose phosphate pathway (PP), including the key enzymes for the oxidative phase involved in anabolic pathways, i.e., glucose-6-phosphate dehydrogenase (Cagg_3190) and 6-phosphogluconate dehydrogenase (Cagg_3189) [23]. Herein, the transcription levels of the two key enzyme genes showed a significant diurnal pattern with the highest relative transcription under high-light conditions (11:00) and an additional peak at 05:00 (Figure 8). Similar to the genes of the oxidative pentose phosphate pathway, high relative transcription levels were also detected at those times for other genes that are indicative of active metabolism and growth, such as DNA gyrase, DNA polymerase, and RNA polymerase (Appendix A). 

Because many of the enzymes involved in glycolysis are bi-directional and similarly used in gluconeogenesis, the transcriptional patterns of genes related to glycolysis were analyzed by focusing on the unidirectional enzyme, 6-phosphofructokinase (Cagg_3643), which irreversibly catalyzes the reaction from fructose-1,6-phosphate to fructose-6-bisphosphate to predict chemoheterotrophic growth and catabolism. In *C. aggregans*, two genes are annotated as genes encoding 6-phosphofructokinase (Cagg_3643 and Cagg_2702). These two genes differ considerably in length, with Cagg_2702 encoding a protein identified as a member of the 6PF1K_euk superfamily, the eukaryotic type of the 6-phosphofructokinase, which is almost twice as long as the ‘bacterial’ 6-phosphofructokinase version, represented by Cagg_3643 (747 aa vs. 356 aa) [64]. Homologs of the Cagg_2702 gene are present in many *Chloroflexota* genomes as identified by a BLAST search, but are not generally present in many other bacteria. The two genes differed in the relative transcription levels and patterns. Cagg_2702 showed the same diel transcription pattern as other genes involved in glycolysis, with its highest relative transcription during high-light conditions at 11:00 (see Appendix A). In contrast, Cagg_3643 showed the highest relative transcription under anaerobic conditions in the evening and the lowest transcription levels during superoxic, high-light conditions (Figure 8), but it showed considerably higher absolute transcription levels (TPM average of 1312.12 vs. 9.28 for Cagg_2702). Because Cagg_3643 represents the ‘bacterial’ type of the enzyme (with higher similarity to the 6-phosphofructokinase in *E. coli*) and since it showed higher absolute transcription levels, it is hypothesized that it represents the unidirectional gene involved in the oxidative activity of glycolysis in *C. aggregans*.

The oxidative TCA cycle is important for oxygen-respiring heterotrophic organisms, and all *Chloroflexus* species are known to have the ability to grow chemoheterotrophically [1,2,3] with oxygen as the terminal electron acceptor. Some of the reactions involved in the TCA cycle—e.g., the conversions from succinyl-CoA to malate—are also part of the 3-OHP bi-cycle [65]. Therefore, the transcriptional patterns of succinyl-CoA synthase (Cagg_2086 and Cagg_2819), succinate dehydrogenase (Cagg_1576–1578), and fumarate lyase (Cagg_2500) are labeled as ‘TCA+3-OHP’ and the genes that are exclusively present in the TCA cycle are labeled as ‘TCA-only’ (Figure 8). Genes involved in the TCA cycle were transcribed relatively evenly over the diel cycle, with two peaks: one in the early morning at 05:00 and the other increasing during the day from 07:00 to 15:00. At 05:00, genes encoding acetate/CoA ligase (Cagg_3789), which catalyzes the production of acetyl-CoA from acetate, also showed significantly higher relative transcription levels (Appendix A). The two TCA-affiliated gene groups showed only small differences in their relative transcription patterns. After a small peak at 06:00 for all TCA-related genes, the relative transcription of the ‘TCA+3-OHP’ genes had already increased at 07:00, whereas the relative transcription levels of the ‘TCA-only’ genes were low at that time and showed a small increase later in the morning (Figure 8).

### 3.8. Transcription of Oxygen Protection Genes

Genes for two oxidative stress-protection enzymes present in the *C. aggregans* genome were analyzed in this study: superoxide dismutase (Cagg_2494) and two copies of a glutathione peroxidase (1: Cagg_0324 and 2: Cagg_0446). Transcripts for the enzymes showed their highest relative transcription levels during the daytime, when both O_2_ and light were present (Appendix A). The gene encoding superoxide dismutase, i.e., an enzyme-detoxifying reactive oxygen species, exhibited a second peak of relative transcripts in the early morning (at 05:00), when cytochrome c oxidase genes were also highly transcribed (see Section 3.4 above, “Phototrophic and Respiratory Electron Transport”). Glutathione peroxidase reduces lipid hydroperoxides to their corresponding alcohols and reduces free hydrogen peroxide to water. Significant high relative transcription levels of glutathione peroxidase-encoding genes in *C. aggregans* were observed at 11:00 (Appendix A), thus indicating the possible presence of not only O_2_ but also hydrogen peroxide.

Hydrogen peroxide is produced as a by-product in the oxidation of glycolate to glyoxylate by glycolate oxidase [66]. The encoding genes (*glcDEF*, Cagg_1528, Cagg_1530–1531, and Cagg_1892–1893) showed patterns similar to that of glutathione peroxidase, with the highest relative transcription levels during midday, indicating both the presence of glycolate in the mat environment and its oxidation by *C. aggregans* (Appendix A). The oxidation of glycolate may be linked to a photoheterotrophic metabolism, which is further supported by the high relative transcript levels for genes encoding glycoside, sugar, and amino acid transporter genes during this time under high-light conditions (Appendix A).

## 4. Discussion

### 4.1. Light and O_2_ Dynamics Shape the Environmental Conditions for C. aggregans

The results of microsensor analyses revealed a strong correlation between solar irradiance and the O_2_ concentration and penetration depth in the cyanobacterial microbial mats from Nakabusa Hot Springs (Figure 2). Since no increase in O_2_ concentration was observed in the bottom layer lower than 1mm in depth, oxygenic cyanobacteria was supposed to be absent in the undermat as supported by the Martinez et al. [13]. The diel changes between anoxia and hyperoxic conditions driven by the oxygenic activity of cyanobacteria lead to drastic changes in the conditions for their microbial metabolism. Consequently, mat-inhabiting microbes may be under optimal conditions during only part of the diel cycle, and may need to endure unfavorable conditions at other times. Under such conditions, a versatile metabolism is thought to be advantageous in terms of ensuring continuous energy production under dynamic environmental conditions. In the following sections it is discussed how *Chloroflexus aggregans* use their metabolic flexibility to thrive in the highly variable conditions in the microbial mat over a diel cycle.

### 4.2. Low Light and Low O_2_ Dominated the Morning Hours (07:00)

After sunrise, although no direct sunlight hit the mats, the irradiance from diffuse light increased and stimulated cyanobacterial oxygenic photosynthesis, as indicated by the increasing O_2_ concentrations at the mat surface as well as the significant increase in the relative transcription levels of genes for protection against reactive oxygen species in *C. aggregans*. However, deeper mat layers were still anoxic, which in combination with low-light conditions seems to provide suitable conditions for anoxygenic photosynthesis by FAPs such as *Chloroflexus spp.* [67]. The increasing transcription of genes encoding housekeeping enzymes suggested increasingly active metabolism (anabolism) in *C. aggregans* (Appendix A). In the morning, the transcription of the phototrophy-affiliated ACIII Cp gene increased slowly and the transcription levels of genes for ACIII Cr, *aurA* and cytochrome *c* oxidase decreased, which in combination indicate active phototrophy. Photoheterotrophy of *C. aggregans* is suggested by increases in the transcription of TCA-related genes, probably using fermentation products in the mats that accumulated during the nighttime, as reported in similar hot spring systems in Yellowstone National Park [68,69,70,71,72]. At the same time, photoautotrophy (and thus photomixotrophy) is indicated by the increased transcription of key 3-OHP genes, probably supported by anaplerotic carbon fixation as indicated by the increased transcription of phosphoenolpyruvate carboxylase genes (Figure 6).

Unexpectedly, neither molecular hydrogen nor sulfide seems to function as an electron donor for autotrophic growth at this time, as neither *hyd* nor *sqr* genes were highly transcribed. In contrast, a significant increase in the relative transcription levels of *cox* gene transcription was seen starting at 07:00, indicating that carbon monoxide is a potential electron source for photoautotrophic metabolism in the early morning, as hypothesized for the CO utilization of *Roseiflexus* spp. and *C. aurantiacus* based on the genomic analysis [11,37].

### 4.3. High-Light and Super-Oxygenated Midday Hours (11:00)

Oxygen started to accumulate in the upper mat layers from around 09:00 and with time also accumulated in the deeper mat layers. Hyperoxic O_2_ levels (>800 µmol O_2_ L^−1^) were observed in the uppermost mat layers and O_2_ penetration reached a >2 mm depth under high irradiance (approx. 1000–1500 µmol photons m^−2^ s^−1^; 400–700 nm) between 10:00 and 14:00. The relative transcription levels of genes encoding DNA gyrase, DNA/RNA polymerase and ATP synthase peaked at the same time (Figure 5 and Appendix A), indicating active growth and energy production of *C. aggregans* during midday high-light and oxic conditions. 

In the laboratory, *C. aggregans* and other FAPs have long been known to grow chemoheterotrophically via oxic respiration under aerobic dark conditions [2]. Additionally, aerobic growth in the light has been shown for *C. aurantiacus* only very recently [73]. In the present study, despite the presence of O_2_ in the upper 2 mm of the mat during this period of the day, there was no indication of chemoheterotrophic growth or aerobic respiration. These results suggest the absence of both active glycolysis (as indicated by low relative transcription of the ‘bacterial’ type of the 6-phosphofructokinase gene) and aerobic respiration (as inferred from the transcription data of cytochrome *c* oxidase genes and the ACIII Cr genes). In contrast, active phototrophy in the presence of O_2_ is suggested by the high relative transcription levels of genes involved in electron transport, including ACIII Cp, *aurB*, and the TCA cycle. Structural analyses indicated that AurA and AurB in *C. aurantiacus* are active during phototrophic growth and chemotrophic growth, respectively [74]. In contrast, the transcription of *aurB* under phototrophic conditions in this study is consistent with the data obtained in a proteomic study of *C. aurantiacus* in which AurA was more abundant during chemoheterotrophic growth, while AurB was observed under photoheterotrophic conditions [45]. While the phototrophic growth under aerobic conditions correlates with earlier laboratory studies demonstrating that O_2_ did not inhibit energy transfer between the chlorosomes and the reaction center in *C. aurantiacus* [75].

However, bacteriochlorophyll biosynthesis in *C. aurantiacus* was long believed to be inhibited by O_2_, and enzymes involved in the biosynthesis were only detected or significantly increased in cultures grown under anaerobic phototrophic conditions [45]. Accordingly, metatranscriptomic studies of alkaline hot spring microbial mats in Yellowstone National Park showed that the transcripts of most of the genes involved in the biosynthesis of BChls in different FAPs, including *Chloroflexus* spp., were the most abundant at night [23]. Inhibitory effects on the biosynthesis of the photosynthetic apparatus under aerobic conditions in *C. aurantiacus* were also shown in the laboratory [76]. In contrast, transcription of phototrophy-related genes under aerobic light conditions were shown for *C. aurantiacus* only very recently [73].

In accordance with the expectation that oxygen represses the biosynthesis of phototrophic apparatus and pigments, nocturnal transcription patterns of the photosynthetic apparatus-related *puf* and *csm* genes were also observed in the present study and might be negatively correlated with the increasing O_2_ concentrations in the environment. In contrast, many of the *bch* genes were observed to be transcribed during the day, with the majority showing the highest relative transcription levels at 11:00. This correlates with a recent study of *C. aurantiacus* showing biosynthesis of BChl *a* and *c* under anaerobic as well as aerobic conditions in the light, while being suppressed in the presence of O_2_ in the dark [73]. This finding supports the hypothesis that *C. aggregans* in Nakabusa Hot Springs cyanobacterial mats can produce BChls for phototrophic growth under aerobic conditions during the daytime.

The high relative transcription levels for glutathione peroxidase-encoding genes during the midday not only reflect high O_2_ levels in the environment; they might also indicate high H_2_O_2_ levels, since the main function of this enzyme is to reduce lipid hydroperoxides to their corresponding alcohols and to reduce free hydrogen peroxide to water [77,78]. Hydrogen peroxide can be formed via photochemical reactions with dissolved organic carbon in hot spring waters [79], and H_2_O_2_ is produced as a byproduct of the oxidation of glycolate (to glyoxylate) [66], which has been shown to be present under high-light conditions in hot spring microbial mats, presumably produced by photoinhibited cyanobacteria [69]. The presence of glycolate and its oxidation at this time of day (11:00) is further suggested by the gene transcription pattern of glycolate oxidase genes, which showed patterns similar to those of the glutathione peroxidase, with highest relative transcription levels during the midday (Appendix A). Glycolate oxidase transcription activity supports the hypothesis that *Chloroflexus* species photoassimilate the glycolate supplied by the cyanobacteria in such microbial mats [80]. This indicates a photoheterotrophic metabolism for *C. aggregans*, which is further supported by the high relative transcript levels for glycoside, sugar, and amino acid transporter genes during high-light conditions (Appendix A).

Carbon monoxide (CO) might function as an electron and/or carbon source under aerobic conditions during this time of day, as aerobic carbon monoxide dehydrogenase genes are found to peak at midday [23]. The transcription profile of carbon monoxide dehydrogenase genes was related to the transcription pattern of the phosphoenolpyruvate carboxylase gene (Figure 6), which indicates that CO might be converted to CO_2_, which then is incorporated by the phosphoenolpyruvate carboxylase-catalyzed anaplerotic reaction (phosphoenolpyruvate to oxaloacetate). In *Thermomicrobium roseum*, an obligately aerobic chemoheterotroph in the phylum *Chloroflexota*, carbon monoxide dehydrogenase is utilized to produce ATP and NADPH under aerobic conditions [81]. It is speculated that *C. aggregans* may use CO for both anaplerotic carbon fixation and supplemental energy production in the presence of O_2_ due to the limitation of available CO_2_ caused by active cyanobacterial photosynthetic carbon fixation.

Two gene clusters encoding the “respiratory” complex I (NADH:menaquinone oxidoreductase, *nuo* genes) are present in *C. aggregans* and *C. aurantiacus* [54,82], as well as in the red FAP species *Roseiflexus castenholzii* and *Roseiflexus* sp. RS-1 (acc. nos. CP000804 and CP000686, respectively [23,37]). To our knowledge, neither of these clusters has been affiliated with phototrophic electron transport. The different relative transcription patterns obtained in the present study might indicate that the 14-gene set is used primarily in photosynthesis and the 12-gene set is used primarily in respiratory electron transport. However, two sets of *nuo* operons have also been described for other, non-phototrophic bacteria such as *Ignavibacterium album* [83] and “*Candidatus* Thermonerobacter thiotrophicus” [18], which might indicate specification to different O_2_ levels rather than phototrophic versus respiratory electron transport. If true, this might indicate a potential higher O_2_ tolerance for the 14-gene set and a higher oxygen affinity for the 12-gene set. Because these results indicate that chemoheterotrophic, respiratory metabolism does not take place during high-light and superoxic conditions at midday, when the 14-gene set is highly transcribed, it is hypothesized that this NADH:menaquinone oxidoreductase plays a role in phototrophic electron transport, perhaps by donating electrons from NADH similar to as it has been proposed to be the case in the cyclic electron transport chain in heliobacteria [84]. However, further biochemical analyses are required to precisely determine the different functions of the two NADH: menaquinone oxidoreductases in *Chloroflexus*.

### 4.4. Low Light and Low O_2_ Dominated the Afternoon Hours (15:00–16:00)

In the afternoon, a substantial decrease in solar irradiance and O_2_ was observed after 14:00, as the sun set behind the surrounding mountains. Between 15:00 and 16:00, O_2_ was still detected only in the upper layer of the mat (Figure 2), and *C. aggregans* is hypothesized to experience microaerobic conditions enabling aerobic chemoheterotrophic metabolism, as indicated by the high relative transcription levels of TCA cycle, ACIII Cr, *aurA*, and cytochrome *c* oxidase-encoding genes (Figure 4, Figure 5 and Figure 8).

Simultaneously, high relative transcription levels of the gene encoding the O_2_-sensitive version of Mg-protoporphyrin monomethylester cyclase, *bchE*, indicated that part of the *C. aggregans* population was exposed to anoxic conditions, at least in the deeper mat layers, as supported by the microsensor data of the vertical O_2_ distribution in the mat (Figure 2 and Figure 3). High relative transcription levels of genes involved in the 3-OHP bi-cycle suggest autotrophic growth, especially under anaerobic conditions, where sulfide and/or H_2_ is available [85]. Sulfide concentrations are expected to rise under anaerobic conditions due to biological sulfate-reduction, as was similarly shown for the bacterial community of Mushroom Spring in Yellowstone National Park [86]. The sulfide oxidation capabilities of *C. aggregans* in the Nakabusa mats are supported by high relative transcription levels of *sqr* for utilization of sulfide as an electron donor [87]. At the same time, uptake hydrogenase genes are transcribed, pointing to the use of molecular hydrogen for autotrophic growth (Figure 7). However, although anoxic low-light conditions were prevalent in the deeper layers of the microbial mats for approx. 2 h before sunset, phototrophy does not seem to be the predominating metabolic growth mode during this time of day, as the decreasing relative transcription levels of ACIII Cp suggest. Thus, sulfide- and H_2_-oxidizing enzymes of *C. aggregans* may be employed for aerobic chemoautotrophic metabolism instead, or additionally for photoautotrophic metabolism at dusk, as suggested by the transcriptional peaks of ACIII Cr, *aurA* and cytochrome *c* oxidase (Figure 4 and Figure 5).

### 4.5. Dark and Anoxic Nighttime Hours (17:00–19:00, 23:00, 02:10)

After 17:00, the microbial mat community experienced dark and anoxic conditions. The low relative transcription levels for housekeeping genes such as DNA gyrase and DNA/RNA polymerases indicate low metabolic activity of *C. aggregans* during this period. The low transcription values for transporter genes support this conclusion. The electron transport gene transcripts for both phototrophy and respiration as well as for the ATP-synthase genes were low. The unidirectional glycolysis enzyme 6-phosphofructokinase gene showed significant changes towards high relative transcription levels at the beginning of the night while TCA cycle-related gene transcriptions were low, suggesting fermentative metabolism and possibly the degradation of internal glycogen storage. This has been shown for *C. aurantiacus* in the laboratory [56] and has been suggested for FAPs inhabiting hot spring microbial mats in Yellowstone National Park [23]. High relative transcription levels of a bi-directional hydrogenase further support the fermentative growth mode and the potential production of H_2_ by *C. aggregans* during the night. This is in accordance with the recent detection of H_2_ production under fermentative conditions (dark, anaerobic) in *C. aggregans* strain NA9-6 (unpublished data).

### 4.6. Early Morning Hours (05:00)

Under dark, anoxic conditions in the early morning hours, an unexpected significant increase in the relative transcription of genes encoding the respiratory chain components (respiratory complexes I, II, and IV and ACIII Cr) as well as ATP-synthase was observed. It is suggested that the transcription of these genes is indicative of the occurrence of chemotrophic growth involving O_2_ respiration at that time of day. Although the microsensor measurements showed no presence of O_2_ in the mats until later in the morning, the transcription of genes encoding enzymes involved in O_2_ protection, such as superoxide dismutase and glutathione peroxidase, indicated a (micro)aerobic environment for *C. aggregans* at that time of day, and trace amounts of O_2_ were detected at the very surface throughout the night. As *C. aggregans* is known to have gliding motility and chemotaxis toward reduced O_2_ concentrations [2,88,89,90,91], it is speculated that *C. aggregans* migrates from anaerobic deeper layers to the micro-oxic surface layers in the early morning, in which a diffusive supply of O_2_ from the overlying water leads to microaerobic conditions during the nighttime, as has been suggested previously [23,92].

Similar to the ACIII Cr operon, the genes of the ACIII Cp operon, presumed to be involved in phototrophy, were also highly transcribed at 05:00. However, active phototrophy is ruled out due to the lack of light. Because the red filamentous anoxygenic phototrophic members of *Chloroflexota*—i.e., *Roseiflexus* spp.—contain only one copy of Cp-like ACIII genes, which are predicted to work under both phototrophic and chemotrophic conditions [23], the Cp-related genes in *C. aggregans* might also function under chemotrophic growth in the mats.

A capacity for chemoautotrophic growth was very recently observed in *Chloroflexus* spp. isolates obtained from Nakabusa Hot Springs microbial mats [8]. The high relative transcription levels of *hyd* uptake hydrogenase and Ni-transporter genes further suggest the use of H_2_ as an electron donor for the aerobic chemoautotrophic growth in the microoxic surface layers of the cyanobacterial mats around 05:00. Additionally, the high transcription levels of genes for the TCA cycle and acetate/CoA ligase—the latter of which catalyzes the production of acetyl-CoA from acetate—indicate that acetate, supplied mainly from the fermentation of co-existing microbes as shown in similar mats in Yellowstone National Park, might be taken up at this time of day [22,68,93,94] (Appendix A). This points to the possibility of an assimilation of acetate in addition to the purely autotrophic metabolism, suggesting the chemomixotrophic lifestyle of *C. aggregans* during predawn.

## 5. Conclusions

This study suggests that *C. aggregans* uses its metabolic flexibility and capability for both phototrophic and chemotrophic growth to optimize its performance under the varying environmental conditions in its natural habitat, the microbial mat community at Nakabusa Hot Springs. The main ATP-generating and thus metabolically most-active times are not only the high-light hours around midday (phototrophy), but—most notably—also the early morning hours around 05:00, when the cells are hypothesized to conduct chemomixotrophic growth (Figure 9).

Genes for the biosynthesis of the photosynthetic apparatus are predominantly transcribed during the night; however, photosynthesis is active during the light hours in the morning, midday and afternoon. Under low O_2_ concentrations in the dim morning light (≤100 µmol photons m^−2^ s^−1^), photoauto/mixotrophic metabolism potentially using CO as an electron donor is suggested to be the major energy source for *C. aggregans* in the cyanobacteria-dominated mats. Later on, under midday high-light conditions, intense oxygenic photosynthesis by cyanobacteria renders the upper millimeters of the microbial mat highly oxic. However, O_2_ respiration in *C. aggregans* does not seem to take place under these conditions. Instead, photoheterotrophic growth (and the assimilation of glycolate) is most likely the dominant lifestyle, supplemented with a certain degree of anaplerotic CO_2_ fixation. In the afternoon, under anaerobic light conditions, photoautotrophic or photomixotrophic growth with sulfide and/or H_2_ as the electron donor takes place in the deeper mat layers, and aerobic respiration and chemoheterotrophic growth are hypothesized for the cells in the upper layers. At nighttime, chemoheterotrophic fermentative growth and the production of H_2_ may take place. In the late-night/early morning hours, at around 05:00, *Chloroflexus* migrates to the mat surface and undergoes mixotrophic growth with H_2_ and O_2_ prior to sunrise, after which *C. aggregans* switches back to phototrophy.

## Figures and Tables

**Figure 1 microorganisms-09-00652-f001:**
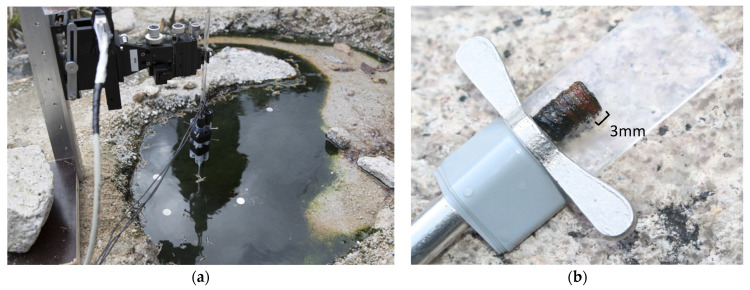
Photographs of the sampling site and cyanobacteria-dominated microbial mat. (**a**) The dark blue-green microbial mats developed at the “Stream Site” of Nakabusa Hot Springs, Japan [19]. (**b**) The microbial mat core samples collected at each time point were 8 mm in diameter and approx. 15 mm thick. The upper 3 mm of the core samples was used in this study.

**Figure 2 microorganisms-09-00652-f002:**
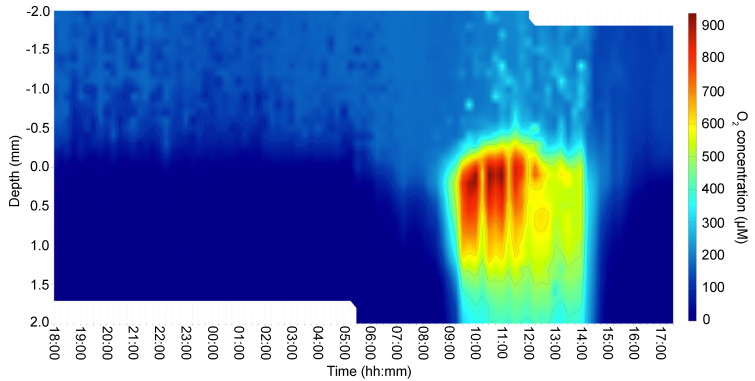
A heat map of the vertical O_2_ concentration profiles in the cyanobacteria-dominated microbial mat of Nakabusa Hot Springs as measured over a diel cycle. The O_2_ concentration (µmol L^−1^) was measured as a function of depth in the microbial mat at 15-min intervals for 24 h from 18:00 on 3 November to 18:00 on 4 November. The mat surface is indicated by 0 mm. Positive depth values indicate the depth below the mat surface, and negative values indicate the depth above the mat surface, i.e., the distance into the overlying water column of the hot spring.

**Figure 3 microorganisms-09-00652-f003:**
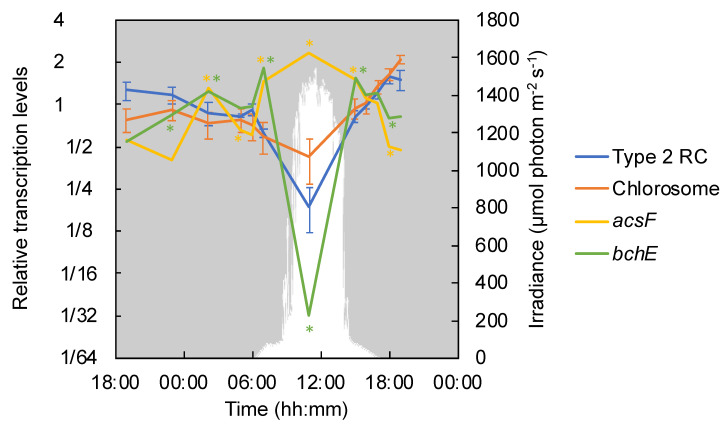
Relative transcription levels of genes encoding photosynthetic reaction center and chlorosome proteins, as well as genes involved in bacteriochlorophyll biosynthesis. Mean values of genes encoding the type 2 reaction center (RC) (*pufLMC*: Cagg_1639–1640 and Cagg_2631) and chlorosome proteins (*csmAMNOPY*: Cagg_1222, Cagg_1209, Cagg_1208, Cagg_2486, Cagg_1206, and Cagg_1296) are represented by a blue line and an orange line, respectively, with standard deviations. The values of Mg-protoporphyrin IX monomethyl ester aerobic cyclase (*acsF*, Cagg_1285, yellow line) and anaerobic cyclase (*bchE,* Cagg_0316, green line) are shown. The downwelling photon irradiance (photosynthetically active radiation [PAR]; 400–700 nm) is indicated in white. The asterisk indicates the transcription of a particular gene corresponding to the color in a timepoint differed significantly (*p* < 0.05) from that in the previous timepoint.

**Figure 4 microorganisms-09-00652-f004:**
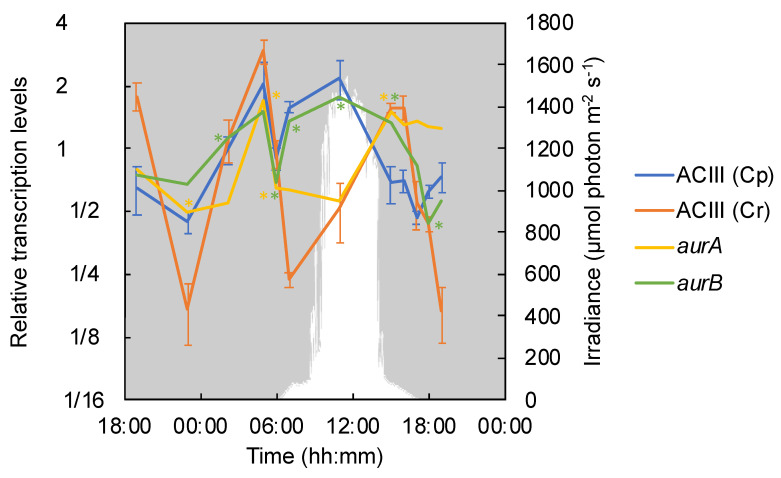
Relative transcriptional levels for alternative complex III (ACIII) and auracyanin. The mean values of ACIII (Cp, Cagg_3382–3383, and 3385–3387) for phototrophic electron transfer and ACIII (Cr, Cagg_1523–1527) for chemotrophic electron transfer are represented by a blue line and an orange line, respectively, with standard deviations. The values of *aurA* and *aurB* (Cagg_0327 and 1833) encoding auracyanin are respectively displayed as yellow (*aurA*) and green (*aurB*) lines. The downwelling photon irradiance (PAR; 400–700 nm) is indicated in white. The asterisk indicates the transcription of a particular gene corresponding to the color in a timepoint differed significantly from that in the previous timepoint.

**Figure 5 microorganisms-09-00652-f005:**
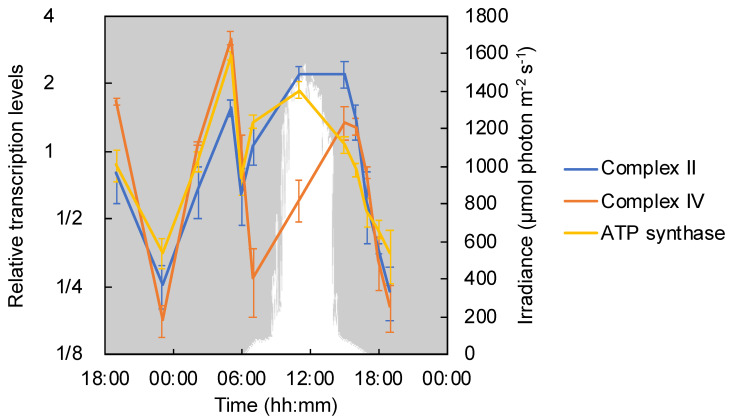
Relative transcription levels of respiratory complex II, IV and ATP synthase. The mean values of the relative transcripts of respiratory complex II (Cagg_1576–1578, blue line), complex IV (Cagg_1519–1522, orange line) and ATP synthase (Cagg_0984—991, yellow line) are shown with standard deviations. The downwelling photon irradiance (PAR; 400–700 nm) is indicated in white.

**Figure 6 microorganisms-09-00652-f006:**
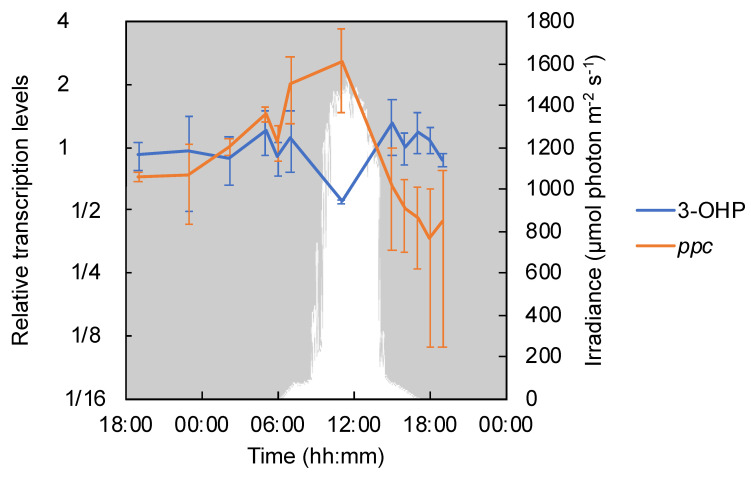
Relative transcription levels of genes encoding key enzymes of the 3-OHP bi-cycle and related enzymes of the anaplerotic pathway in cyanobacterial mats. The mean values of the relative transcription levels of key 3-OHP enzymes, i.e., malonyl-CoA reductase (Cagg_1256) and propionyl-CoA synthase (Cagg_3394), plus that of phosphoenolpyruvate carboxylase (*ppc*, Cagg_0058 and 0399) are represented by a blue line and an orange line, respectively, with standard deviations. The downwelling photon irradiance (PAR; 400–700 nm) is indicated in white.

**Figure 7 microorganisms-09-00652-f007:**
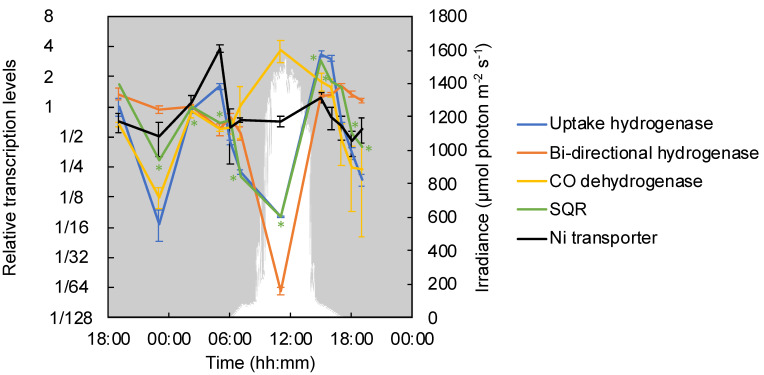
Relative transcription levels of genes encoding hydrogenases, sulfide:quinone oxidoreductase and nickel transporter in cyanobacterial mats. The mean values of the relative transcription levels for type-I uptake Ni-Fe hydrogenase genes *hydAB* (Cagg_0470–0471, blue line), bi-directional Ni-Fe hydrogenase genes homologous to *frhA*, *frhG*, *hoxU*, *nuoF* and *hoxE* (Cagg_2476–2480, orange line), carbon monoxide (CO) dehydrogenase genes *coxGSML* (Cagg_0971–0974, gray line), and nickel transporter genes (Cagg_1273–1276, light blue line) are shown with standard deviations. The relative transcription levels of the type-II sulfide:quinone oxidoreductase gene (*sqr*, Cagg_0045) are represented by the black line. The downwelling photon irradiance (PAR; 400–700 nm) is indicated in *white*. The asterisk indicates the transcription of a particular gene corresponding to the color in a timepoint differed significantly from that in the previous timepoint.

**Figure 8 microorganisms-09-00652-f008:**
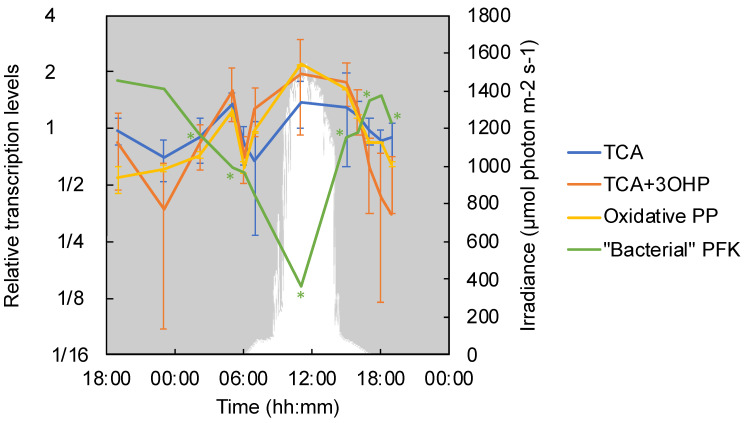
Relative transcription levels of genes for carbohydrate metabolism. The mean values of the relative transcriptional levels for the TCA cycle (Cagg_3738, 3721, 2500, and 2290) and common enzymes of the TCA cycle and the 3-OHP bi-cycle that is labeled as ‘TCA+3-OHP’ (Cagg_2086, 2819, and 1576–1578) are represented by a blue line and an orange line with standard deviations. The mean values of the relative transcripts of genes encoding key enzymes of the pentose phosphate pathway (Oxidative PP)—i.e., glucose-6-phosphate dehydrogenase (Cagg_3190) and 6-phosphogluconate dehydrogenase (Cagg_3189)—are represented by the yellow line with standard deviation. The values of the relative transcription levels are displayed for genes encoding the ‘bacterial’ type of the 6-phosphofructokinase that is labeled as “Bacterial” PFK (Cagg_3643, green line). The downwelling photon irradiance (PAR; 400–700 nm) is indicated in white. The asterisk indicates the transcription of a particular gene corresponding to the color in a timepoint differed significantly from that in the previous timepoint.

**Figure 9 microorganisms-09-00652-f009:**
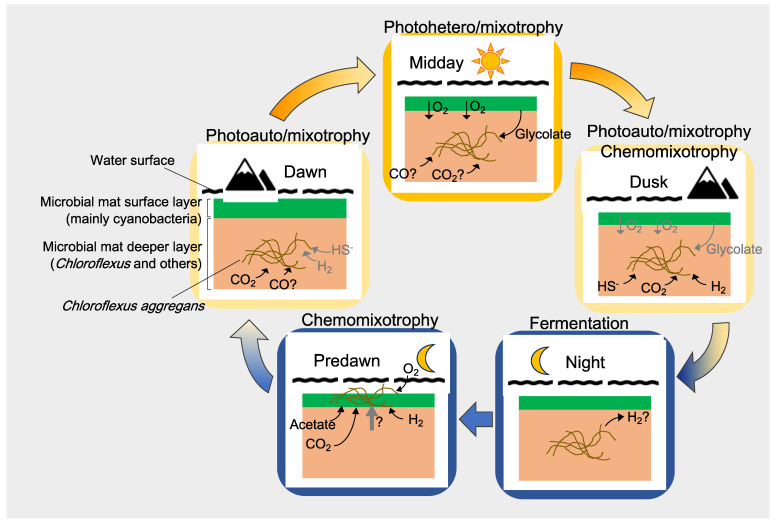
Proposed diel growth modes of *C. aggregans* (indicated by brown filaments) in cyanobacterial mats based on in situ metatranscriptomic and microsensor analyses in the cyanobacterial mats of Nakabusa Hot Springs. The green area represents the green upper layer containing oxygenic phototrophs (i.e., cyanobacteria), while the orange area corresponds to the orange colored undermat. The black curvy line indicated the overflowing water surface.

## Data Availability

The sequence collected in this study is available under NCBI BioProject accession number PRJNA715822.

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
