# Peer review of "In-Situ Metatranscriptomic Analyses Reveal the Metabolic Flexibility of the Thermophilic Anoxygenic Photosynthetic Bacterium Chloroflexus aggregans in a Hot Spring Cyanobacteria-Dominated Microbial Mat"

_microorganisms, 2021, doi:10.3390/microorganisms9030652_

Round 1

Reviewer 1 Report

The manuscript by Kawai, Thiel, and coworkers, “In-situ metatranscriptomic analyses reveal the metabolic flexibility of the thermophilic anoxygenic photosynthetic bacterium Chloroflexus aggregans in a hot spring cyanobacteria-dominated microbial mat” (microorganisms-1108447), describes the detailed and meaningful in situ study of a photosynthetic filamentous bacterium that plays an important environmental role in the cyanobacterial dominated microbial mats of Nakabusa Hot Springs. There exists relatively little earlier literature about the topical organism, Chloroflexus aggregans, and this manuscript will represent a marked contribution to the field. The methods employed by this research are sound, and the implications of the results may be of widespread significance to the readership. I am suggesting that this manuscript be accepted for publication after minor revisions, with specific or broad suggestions for improvement that follow:

1) There is an apparent discrepancy between the abstract LL 30-33 and the discussion LL 685-694. The abstract remarks that this is the first report of C. aggregans as a chemoautotroph, but reference 8 and the discussion would imply otherwise. Is this organism chemoautotrophic or not? Has it been appropriately reported in the past or not? If the authors wish to contest the conclusions of reference 8, then it should be done explicitly and conclusively rather than implicitly and leaving it to the reader’s imagination. In any case, the text on LL 30-33 and LL 685-694 should be brought into a clear alignment, which should further be made to match what is presented in the graphic of Figure 9.

2) Refs 13 and 24, and the discussion on LL 72-75 present the relative abundance of C. aggregans in the Nakabusa Hot Spring cyanobacterial mats as “3%-31%” – which is a reasonable range, but also quite broad. Was the relative abundance of the organism tested for the samples otherwise evaluated in the present study? Was the same measurement made across different layers of the microbial mat? Refs 13 and 24 share much of the authorship with his manuscript, so the capability would appear to be present.

3) The experimental section regarding the microsensor analyses (LL 144-155) is slightly lacking. Presented on line 153 is the conceptual definition of the mat surface as 0 μm relative to the negative depths above that surface and positive depths below that surface where oxygen measurements were taken. It should be indicated what number of measurements were collected per depth, as well as at how many depths measurements were made… or the depth distance that the microsensor tip was maneuvered between measurements. It is clear from Figure 2 that many measurements were made at different depths, but at present there remains some obscurity to this information.

4) While interesting and attractive, Figure 1 does not have an obvious impact on the manuscript body text at which it is referenced (line 167). I recommend rewording or adding a sentence to explain the inclusion of this figure.

5) Further regarding Figure 1, the graphic in panel A is of relatively poor quality or likely just image resolution, especially compared to the crisp quality graphic shown in panel B. I suggest Figure 1A be replaced with an improved image file.

6) Regarding Figure 2, how does this depth-to-oxygen analysis correlate with the cell density or relative abundance of C. aggregans in the core samples with respect to cyanobacteria or algae? What proportion of the photosynthesis indicated here might be attributed to C. aggregans? I think these points merit further discussion in the text.

7) On a related note, I think it would be a valuable addition for the authors to include a brief discussion about the difference between photosynthetic machinery (genes, proteins, chlorophylls) in cyanobacteria and C. aggregans to help the reader to better understand the specificity of the metatranscriptomics analysis of this study as it relates explicitly to C. aggregans.

8) For a point about style, Figure S2 is referenced in the text (i.e. line 263) prior to the first mention of Figure S1 (line 408). Perhaps the editor can offer some insight into the acceptability of this practice given the supplementary nature of the figure in question, but I find it confusing as a reader. I suggest renumbering of the supplementary figures to match the order of their calling references in the manuscript, and careful alignment of the manuscript text and supplementary materials description (LL 727-731) to prevent introduction of any new errors by doing so.

9) I strongly feel that Figure 9 should be redrawn to meet the potential it holds.

a) For one thing, the text font and/or spacing between characters in the same words is quite unattractive and distracting. If this is an issue with the generation of the pdf version, the whole figure including text could be prepared and saved as an image file for high-resolution insertion.

b) Another issue is that most of the text labels are so small (i.e. for the elemental factors) that they are not legible at 100% scale on the screen and are hard to read in print.

c) The indication of sunrise and sunset by a sun on either side left or right behind the same mountain is confusing. I suggest that the sun and mountains of ‘dawn’ could be moved to the left side of that panel, while the sun and mountains of dusk could remain in place – thus giving the appearance of a southward perspective on the hot spring and microbial mat with different mountains on either side.

d) “OC” on the left side of the predawn and dusk panels is really not a suitable way to present CO.

e) “Surface layer” in the dawn panel should read “Microbial mat surface layer” and

f) “Chloroflexus” as well as “Chloroflexus aggregans” in the dawn panel should be italicized.

Reviewer 2 Report

L77-86: The 'we' form, in my opinion, should not be used in scientific papers; the impersonal form is more correct. The same remark applies to the whole chapters Discussion and Conclusions.

L88-106: were samples taken in replicates? Was RNA isolation then performed from each sample in the sense in replicates, or from pooled samples?

L93: what does the notation "#4" before cork borer mean? Size? Please be more specific.

L116-129: any references?

L126: Agilent Technologies, Palo Alto, CA.

L148, 150: Unisense, Aarhus, Denmark.

L160: no statistical analysis of the results obtained? Then how can you compare the results and talk about changes?

Figure 1: this figure should be placed in the methodology, not the results.

Figure 2: I suggest enlarging the font for the legend and the axes, at the moment it is a bit unreadable.

L221: AcsF and BchE should they not be written in italics?

L217-231: Here there is a form of discussion and references to literature, I would move this passage to discussion or reword it.

Figure 3 and 6: It would be helpful if the graphics were larger. It would also be worth increasing the font size and changing it to a standard font, as it is not consistent with the rest of the paper. The legend should be placed next to the graph. Shouldn't the X axis have a unit of some sort? What are vertical lines?

L241-289: Here there is a form of discussion and references to literature. It does not fit into the results section.

Figure 4, 5, 7. 8: It would be helpful if the graphics were larger. It would also be worth increasing the font size and changing it to a standard font, as it is not consistent with the rest of the paper. Shouldn't the X axis have a unit of some sort? What are vertical lines?

L306-320: Here there is a form of discussion and references to literature. It does not fit into the results section.

L326-346: Here there is a form of discussion and references to literature. It does not fit into the results section.

L400-441: Here there is a form of discussion and references to literature. It does not fit into the results section.

L489-500; 527-533; 620-631 – this is the presentation of the results.

L702-726: A short 1-2 sentence conclusion is missing. Should not refer to figures, tables in conclusion (L708).

In general, I think that the results section needs a lot of reconstruction. At the moment there are many discussion elements in it. The results should present briefly and concretely the data obtained. Any discussion of them, explaining the reasons for one result and not another, and a comparison with the literature should be included in the discussion. At the moment, these two chapters are mixed up. The results, in my opinion, lack statistical analysis. The graphs do not really tell us whether the differences obtained are statistically significant.

Round 2

Reviewer 2 Report

Following on my earlier comments and your answers:

1) L88-106: were samples taken in replicates? Was RNA isolation then performed from each sample in the sense in replicates, or from pooled samples?

A: Samples were taken in triplicates. RNA was extracted and sequenced from a single replicate since the RNA concentrations obtained were sufficient from each of the samples.

And

L160: no statistical analysis of the results obtained? Then how can you compare the results and talk about changes?

A: The reviewer is correct. We did not conduct statistical analysis of the transcription values obtained. As only one single RNAseq run (either of a single or pooled RNA extract) was conducted for each timepoint, true statistics were not possible. In an ideal world the data would have been obtained in replicates, both biological and technical. Unfortunately, limitations in both the amount of biomass available as well as the monetary funding prevented this. However, we conducted normalization of the counts and obtained data from a set of 12 samples of the same microbial mat; the samples of adjacent timepoints qualify as the most important ‘controls’ for the data. True replicates would be difficult to obtain as even samples of the same timepoint don’t necessarily contain the same amount of Chloroflexus cells. Another ‘control’ is the comparison to other studies on similar organisms. We discuss the obtained results in comparison to Chloroflexus spp. In similar hot spring microbial mats in the USA in the manuscript. Further, this approach has been shown to be suitable for the detection of transcriptional patterns even without additional statistical analysis as shown by previous studies (e.g., Thiel et al. 2019, Klatt et al. 2013, Liu et al. 2012).

  • R2: Then, in my opinion, such information should be included in the paper. E.g. in the discussion add a subsection "limits" or something like that. I understand financial limitations, but such a large number of results calls for statistical analysis and the reader is a bit surprised (I was) not to find it in the paper. I recommend to include clear information on how many samples there were, what was done in replicates and what was not and why we do not have statistical analysis. Such information shows the authors' awareness that the work is not perfect, and this does not in any way detract from the scientific level of the research.

2) L217-231: Here there is a form of discussion and references to literature, I would move this passage to discussion or reword it.

A: The authors agree that the results section(s) contains references to literature. This might be not strictly in the scope of a results section. However, this information should be placed here with the intention to guide the reader and allow a better understanding of the methodical setup and the chosen genes analyzed. A strict adherence to the ‘results never contain references to literature, but only data’ dogma was perceived as hindering for both reading the text and following the author’s thoughts. Whereas, the introduction and inclusion of some theoretical background was perceived as helping it. However, we have carefully revised and removed all unnecessary and redundant parts of e.g., the gene introductions from the results part in order to comply to the reviewer’s comment and shorten the section.

  • R2: Thank you for your answer and for correcting the text. As far as the introduction of information explaining the methodical setup and the chosen genes analysed is concerned, I think it should be in the methodology. It would make more sense.

3) L295, 394, 446, 554, 623, 632, 683, 727 (in the revised version): the form 'we', 'our' reappears, although this has been corrected elsewhere in the text.

Summary, the discussion and results now look better. The figures are clear. I appreciate the corrections made and, apart from the above three comments, I have no objections.
